# Solvent-Controlled Morphology of Zinc–Cobalt Bimetallic Sulfides for Supercapacitors

**DOI:** 10.3390/molecules28186578

**Published:** 2023-09-12

**Authors:** Xiaobei Zang, Xiaoqi Tang, Liheng Liang, Xuhui Liu, Xiaobin Zhang, Xingdong Ma, Guoshun Liu, Chao Li, Ning Cao, Qingguo Shao

**Affiliations:** State Key Laboratory of Heavy Oil Processing, School of Materials Science and Engineering, China University of Petroleum (East China), Qingdao 266580, Chinaz22140037@s.upc.edu.cn (G.L.);

**Keywords:** supercapacitor, bimetallic sulfide, morphology control, ZnCoS

## Abstract

Bimetallic sulfides offer high theoretical specific capacitance and good stability as electrode materials due to their diverse redox reactions, larger specific surface areas, and better conductivity. The morphology of the electrode material is an important influencing factor for the electrochemical properties. Herein, a series of ZnCoS electrode materials with different morphologies were prepared by varying the solvent in the solvothermal reaction, and the effects of different microstructures on the electrochemical properties of ZnCoS were investigated. The ratio of water and ethanol in the solvent was controlled to modulate the microstructure of the as-prepared ZnCoS materials. XRD and XPS revealed the physical and chemical structure of the ZnCoS materials. SEM and TEM observations showed that the microstructure of ZnCoS transformed from one-dimensional wires to two-dimensional sheets with increasing amounts of ethanol. The maximum specific capacitance of the as-prepared ZnCoS materials is 6.22 F cm^−2^ at a current density of 5 mA cm^−2^, which is superior to that of most previously reported bimetallic sulfides. The enhanced electrochemical performance could be ascribed to its sheet-assembled spherical structure, which not only shortens the path of ion diffusion but also increases the contact between surface active sites and the electrolyte. Moreover, the spherical structure provides numerous void spaces for buffering the volume expansion and penetration of the electrolyte, which would be favorable for electrochemical reactions. Furthermore, the ZnCoS electrodes were coupled with activated carbon (AC) electrodes to build asymmetric supercapacitors (ASCs). The ASC device exhibits a maximum energy density of 0.124 mWh cm^−2^ under a power density of 2.1 mW cm^−2^. Moreover, even under a high-power density of 21 mW cm^−2^, the energy density can still reach 0.055 mWh cm^−2^.

## 1. Introduction

With the rapid increase in the world population, the demand for energy has been continuously rising, and the extensive use of fossil fuels has resulted in a series of environmental problems [1]. Therefore, searching for a safer, green and efficient energy system to replace traditional fossil energy is an inevitable choice for human social development [2]. Supercapacitors (SCs) are a kind of energy storage device with high-power density, and their excellent safety and cycling stability have attracted widespread attention [3,4]. However, compared with batteries, SCs have lower energy density, which limits their practical applications [5]. Generally, supercapacitors can be classified into two types based on the charge storage mechanism: electrical double layer capacitors (EDLCs) and pseudocapacitors (PCs) [6]. EDLCs store energy by the accumulation of ions at the electrode–electrolyte interface [7]. PCs store energy through reversible faradaic reactions of ionic species [8], and they exhibit higher specific capacitance and higher energy density. The electrode material of supercapacitor is the main factor determining its performance, and therefore, numerous efforts have been made to develop advanced electrode materials for better electrochemical performance [9].

Recently, transition metal sulfides (TMSs) have received significant attention as pseudocapacitive electrode materials due to their high theoretical specific capacity, diverse types, and environmental friendliness [10,11]. Compared to the corresponding oxides, TMSs exhibit better electrochemical performance and excellent conductivity [12,13]. At present, monometallic sulfides have been intensively studied, and electrode materials such as CoSx [14], NiSx [15], CuS [16], WS_2_ [17], and MoS_2_ [18] with different morphologies and structures have been reported. Compared to monometallic sulfides, bimetallic sulfides offer higher theoretical specific capacitance due to their diverse redox reactions [19,20]. Additionally, bimetallic sulfides possess better stability, larger specific surface area, and better conductivity, providing more opportunities to enhance the performance of electrode materials [21,22]. Currently, TMSs such as NiCo_2_S_4_ [23], ZnCoS [24], and CuCo_2_S_4_ [25] have demonstrated excellent electrochemical performance. However, bimetallic sulfides suffer from volume effects and slow kinetics during electrochemical reactions, resulting in poor cycling stability and rate capability [26].

PCs rely on redox reactions occurring on or near the electrode surface to store charge [27], so their performance is closely related to the morphology of the electrode material. Effective morphology engineering can shorten the diffusion path of ions, thereby improving the rate performance of electrode materials [28]. It can also increase the specific surface area of the electrode material, providing a higher specific capacitance. Selecting precursor materials with specific morphologies is an effective approach to obtain three-dimensional nanostructures. Guo et al. [29] synthesized NiCo-MOF by replacing 1,4-dicarboxybenzene with trimesic acid, resulting in a composite structure of microspheres and nanosheets. After vulcanization, the NiCoS-2 electrode exhibited a maximum specific capacitance of 1790.4 F g^−1^ and demonstrated excellent electrochemical performance. Varying a specific parameter in the reaction system to obtain a series of microstructures with different morphologies is also an effective method for morphology control. Wu et al. [30] synthesized hierarchical NiS/carbon hexahedrons formed by self-assembling nanoplates or nanorods using the nitrilotriacetic acid (NTA)-assisted hydrothermal method. By adjusting the concentration of NTA, the micromorphology of the hexagonal unit structure could be controlled, thus optimizing the electrochemical performance of the electrodes.

The solvent plays an important role in the wet-chemical synthesis of inorganic nanomaterials, and tuning the solvent is an effective strategy for controlling the microstructure of products. Therefore, in this study, a series of ZnCoS electrode materials with different morphologies were prepared by varying the solvent in the solvothermal reaction, and the effects of different microstructures on the electrochemical properties of ZnCoS were investigated. ZnCoS nanostructures were grown on Ni foam by solvothermal methods, and the ratio of water and ethanol in the solvent was controlled to modulate the microstructure of the as-prepared ZnCoS materials. It was found that the microstructure of ZnCoS transformed from one-dimensional wires to two-dimensional sheets with increasing amounts of ethanol. The maximum specific capacitance of the as-prepared ZnCoS materials is 6.22 F cm^−2^ at a current density of 5 mA cm^−2^, which is superior to that of most previously reported bimetallic sulfides. The enhanced electrochemical performance could be ascribed to its sheet-assembled spherical structure, which not only shortens the path of ion diffusion but also increases the contact between surface active sites and the electrolyte. Moreover, the spherical structure provides numerous void spaces for buffering the volume expansion and penetration of the electrolyte, which would be favorable for electrochemical reactions. Furthermore, the ZnCoS electrodes were coupled with activated carbon (AC) electrodes to build asymmetric supercapacitors (ASCs). The ASC device exhibits a maximum energy density of 0.124 mWh cm^−2^ under a power density of 2.1 mW cm^−2^. Moreover, even under a high-power density of 21 mW cm^−2^, the energy density can still reach 0.055 mWh cm^−2^.

## 2. Results and Discussion

### 2.1. Morphology and Structure Analysis

The preparation process of ZnCoS electrode materials is shown in Figure 1a. The zinc–cobalt precursor was grown on the surface of nickel foam by the solvothermal method. Subsequently, the ZnCoS electrode material was obtained by vulcanization of the zinc–cobalt precursor. The phase structure and crystallinity of the ZnCoS samples were investigated by X-ray diffraction (XRD), as shown in Figure 1b. The diffraction peaks at 2θ = 28.6°, 33.2°, 47.6° and 56.5° correspond to the (111), (200), (220) and (311) crystal planes of Zn_0.76_Co_0.24_S (PDF 47-1656), indicating the successful preparation of Zn_0.76_Co_0.24_S under different ethanol ratios [31]. ZnCoS-H_2_O, prepared using water as the solvent, exhibits the sharpest diffraction peaks, suggesting better crystallinity of the sample [32]. To analyze the influence of different solvents on the crystallinity of ZnCoS, the average grain size of the samples (K = 0.94) was calculated using the Scherrer equation based on the full width at half maximum (FWHM) of the three prominent diffraction peaks at 2θ = 28.6°, 47.6°, and 56.5° [33]. The average grain sizes of ZnCoS-H_2_O, ZnCoS-2:1, ZnCoS-1:1 and ZnCoS-1:2 were determined to be 20.8 nm, 12.2 nm, 14.3 nm and 14.6 nm, respectively. The results demonstrate that the addition of ethanol did not affect the phase structure of the samples, but it decreased the crystallinity and grain size of ZnCoS.

The elemental composition and valence states of the ZnCoS-1:2 surface were analyzed by X-ray photoelectron spectroscopy (XPS). The XPS full-survey spectrum reveals the presence of C, O, Co, Zn and S, as shown in Figure 2a. In Figure 2b, two prominent peaks at 1021.8 eV and 1045 eV correspond to Zn 2p_3/2_ and Zn 2p_1/2_, respectively, indicating the presence of Zn^2+^ in the sample [34]. The high-resolution Co 2p spectrum (Figure 2c) consists of two spin-orbit peaks and two shake-up satellite (Sat.) peaks. The Co^3+^ states are represented by peaks at 779.1 eV and 797.7 eV, while the Co^2+^ states are represented by peaks at 781.6 eV and 800.1 eV, confirming the presence of mixed Co^3+^/Co^2+^ states [35]. The S 2p spectrum (Figure 2d) can be deconvoluted into two peaks corresponding to S 2p_3/2_ and S 2p_1/2_ [36]. The XPS results show that Zn, Co and S exist in the forms of Zn^2+^, Co^3+^, Co^2+^ and S^2−^, respectively, which is in good agreement with the results of the phase structure analysis.

Scanning electron microscopy (SEM) was used to characterize the morphology evolution of ZnCoS samples prepared under different ethanol ratios. Figure 3a shows the morphologies of the ZnCoS-H_2_O sample on Ni foam; it displays a smooth surface in the central area and aggregated particles in the edge area. Figure 3b shows the enlarged SEM image of the smooth area, which reveals a one-dimensional wire structure. Figure 3c is the enlarged SEM image of the edge area, which exhibits a wire-assembled flower structure. When a small amount of ethanol was added to the solvent (the percentage of ethanol is 33%), the morphologies of ZnCoS-2:1 (Figure 3d–f) were similar to those of ZnCoS-H_2_O (Figure 3a–c), which consisted of one-dimensional nanowires. Compared with ZnCoS-H_2_O, there was a notable difference at the edge of the nickel foam scaffold where the orderliness of the spherical structure of ZnCoS-2:1 decreased, consisting of randomly interconnected nanowires. When the ratio of ethanol to water was increased to 1:1 (the percentage of ethanol is 50%), the microstructure of ZnCoS-1:1 (Figure 3g–i) was transformed into two-dimensional sheet structures. As the ethanol concentration in the solution further increased to 67%, the area of ZnCoS-1:2 (Figure 3j–l) nanosheets increased, accompanied by an improvement in orderliness, and the flower-like structure at the edge of the skeleton transformed into a sheet-assembled spherical structure. This porous structure not only shortens the path of ion diffusion but also increases the contact between surface active sites and the electrolyte. Moreover, the spherical structure provides numerous void spaces for buffering volume expansion and penetration of the electrolyte, which would be favorable for electrochemical reactions. In addition, the elemental mappings of the uniformly distributed Zn, Co and S elements exhibited in Figure 3m–o confirm the successful synthesis of the ZnCoS bimetallic sulfides on the surface of NF.

The morphology changes can be explained by the interaction between the surfactant CTAB and ethanol. With an increase in ethanol content, the dielectric constant of the solution decreases, and the polarity weakens while the solubility of the surfactant increases. The structure-oriented effect of CTAB on ZnCoS growth is weakened, and the morphology of ZnCoS changes from one-dimensional to two-dimensional. Figure 4 displays the transmission electron microscopy (TEM) image and the high-resolution TEM (HRTEM) image of ZnCoS-1:2. The TEM image (Figure 4a) reveals that the sheet consists of nanoparticles with diameters ranging from 10 to 30 nm. The lattice fringes in the HRTEM image (Figure 4b) are in good agreement with the (111) crystal face of Zn_0.76_Co_0.24_S.

### 2.2. Electrochemical Characterization

To explore the influence of different ethanol concentrations on the electrochemical performance, electrochemical tests were measured using a three-electrode system in a 6 M KOH aqueous electrolyte. Figure 5a shows the cyclic voltammetry (CV) curves of ZnCoS-H_2_O, ZnCoS-2:1, ZnCoS-1:1, and ZnCoS-1:2 electrodes at a scan rate of 5 mV s^−1^. All CV curves exhibit strong oxidation peaks in the potential window range of 0.35–0.55 V and strong reduction peaks in the potential window range of 0.15–0.35 V, indicating the pseudocapacitive behavior of the electrode materials [37]. The reversible faradaic reaction occurring on the working electrode can be represented as [38,39]:(1)$$ZnS+{OH}^{-}⇌ZnSOH+e^{-}$$
(2)$$CoS+{OH}^{-}⇌CoSOH+e^{-}$$
(3)$$CoSOH+{OH}^{-}⇌CoSO+H_{2}O+e^{-}$$

Among all the samples, the CV curve of ZnCoS-1:2 shows the largest integrated area and highest current density, demonstrating that it has a larger capacitance. To further calculate the specific capacitance and understand the rate capability of the ZnCoS electrodes prepared in different ethanol concentrations, galvanostatic charge–discharge (GCD) tests were carried out. Figure 5b shows the GCD curves of ZnCoS-H_2_O, ZnCoS-2:1, ZnCoS-1:1, and ZnCoS-1:2 at a current density of 5 mA cm^−2^; the voltage plateaus are consistent with the CV results. As can be observed, the ZnCoS-1:2 electrode exhibits the longest discharge time, proving the highest specific capacitance, consistent with the conclusion from the CV tests [40]. Figure 5d illustrates the specific capacitance under different current densities. When the current density is 5 mA cm^−2^, the specific capacitances of the ZnCoS-2:1, ZnCoS-1:1 and ZnCoS-1:2 electrodes are 5, 3.66 and 6.22 F cm^−2^, respectively. ZnCoS-1:2 exhibits a larger specific capacitance compared to previously reported bimetallic sulfides such as ZCS@ZCNS [41], CNTF@ZCO [42] and NiCo_2_S_4_ [43]. In addition, CV curves under different scanning rates and GCD curves under different current densities of ZnCoS-1:2 are shown in Figure 5e,f, respectively. The little shift in the anodic or cathodic peak in CV curves and stable discharge plateaus demonstrate the low polarization and good rate stability of ZnCoS-1:2 electrodes. The enhanced electrochemical performances should be ascribed to the morphological features of ZnCoS-1:2, in which the continuous sheet-assembled structure facilitates faster charge and ion transfer, resulting in a higher specific capacitance. Figure 5c presents the electrochemical impedance spectroscopy (EIS) plots of the electrodes. The equivalent series resistance (ESR) reflects the conductivity of the electrodes, and the contact between the current collector and the electrodes also affects it [44,45,46]. The value of ESR is determined by the intercept on the x axis. As shown in Figure 5c, the ESR values of ZnCoS-2:1, ZnCoS-1:1 and ZnCoS-1:2 are 0.67, 0.57 and 0.68 Ω, respectively, indicating a strong connection between ZnCoS and the Ni foam current collector. In the low-frequency region, the slopes of the three lines are similar, indicating the ideal capacitance behavior. In order to further investigate the effect of morphological changes on the electrochemical behavior of ZnCoS, the charge storage mechanism and reaction kinetics of each electrode were investigated. Figure 5g–i display the relationship between the log (scan rate) and log (current peak). The equation for the relationship is as follows [47]:(4)$$i=av^{b}$$
(5)log⁡(i)=b log(v)+log(a)
where *i* refers to the peak current density, *v* refers to the scan rate, and *a* and *b* are parameters. In general, the electrochemical kinetics are controlled by a diffusion-controlled process when *b* = 0.5 and a surface-controlled process when *b* = 1. Calculations show that the *b*-values of all three electrode materials, ZnCoS-2:1, ZnCoS-1:1, and ZnCoS-1:2, are close to 0.5, which implies that diffusion control dominates the electrochemical reactions in the redox peaks, which is consistent with the properties of the battery-type pseudocapacitive electrode materials [48,49]. The b-value of ZnCoS-1:2 is calculated closer to 0.5 than that of ZnCoS-2:1 and ZnCoS-1:1, which is attributed to the fact that the crosslinked two-dimensional nanosheets can shorten the transfer path of the ions and electrons, thus increasing diffusion in the charge storage process and promoting the electrochemical performance.

To evaluate the practical application, the two-electrode asymmetric supercapacitors (ASC) were constructed using ZnCoS-1:2 as the positive electrode and activated carbon (AC) as the negative electrode. Figure 6a shows the CV curves of ZnCoS-1:2 and the AC electrode in the KOH electrolyte, which suggests a well-matched electrochemical window of the two electrodes. Figure 6b shows the CV curves of the ZnCoS-1:2//AC ASC at different scan rates. All curves have similar shapes with little polarization under various scan rates, indicating that the ASC has good electrochemical activity. The typical GCD curves (Figure 6b) demonstrate good symmetry with the potential time curves at different current densities (3–30 mA cm^−2^), reflecting good reversibility of the devices. When the current density is 3 mA cm^−2^, the specific capacitance of the ZnCoS-1:2//AC ASC is 454 mF cm^−2^. Figure 6d presents the EIS plot of ZnCoS-1:2//AC ASC. The device exhibits low charge transfer resistance and internal resistance (1.963 Ω and 1.66 Ω, respectively), which guarantees its high specific capacitance.

In addition, the ZnCoS-1:2//AC ASC exhibits a good rate of performance and cycling stability. As the current increases ten times from 3 mA cm^−2^ to 30 mA cm^−2^, the specific capacitance decreases from 486 mF cm^−2^ to 203 mF cm^−2^, with a capacitance retention of 44.5% (Figure 7a). Figure 7b shows the Ragone plot of ZnCoS-1:2//AC ASC. As shown in Figure 7b, the device exhibits a maximum energy density of 0.124 mWh cm^−2^ under a power density of 2.1 mW cm^−2^. Moreover, even under a high-power density of 21 mW cm^−2^, the energy density can still reach 0.055 mWh cm^−2^. Such excellent electrochemical performances are superior to some previously reported devices such as NiCo_2_S_4_//C [43], rGOFF-MoS_2_//rGOFF-MoS_2_ [50], PPy-hs@CoS//AC [51], and MnO_2_/CNT//PI/CNT [52]. In addition, a long-term galvanostatic charge–discharge test was also performed to evaluate the cycling stability of the device, as shown in Figure 7c. The ZnCoS-1:2//AC ASC retained 68.4% of the initial capacitance after 5000 cycles at a current density of 30 mA cm^−2^, indicating good cycling stability. Considering the high specific area of normalized capacitances, large energy density and good cycling performance, the assembled ZnCoS-1:2//AC ASC has a potential application as an energy storage device in the future.

## 3. Experimental

### 3.1. Chemicals and Materials

Hydrochloric acid, acetone, ethanol, Zn(NO_3_)_2_·6H_2_O, and Na_2_S·9H_2_O were purchased from Sinopharm Chemical Reagent Co., Ltd., Shanghai, China. Co(NO_3_)_2_·6H_2_O, urea, and cetyltrimethylammonium bromide (CTAB) were purchased from Shanghai Macklin Biochemical Co., Ltd. (Shanghai, China). The above reagents were used directly without further purification. The nickel foam (NF) was purchased from Taiyuan Lizhiyuan Battery Material Co., Ltd. (Taiyuan, China). Before use, the NF was cut to 1 × 2 cm, cleaned with acetone, ethanol, 1 M hydrochloric acid, deionized water and ethanol for 15 min in turn under ultrasound conditions, and dried in a vacuum drying oven at 60 °C for 12 h.

### 3.2. Material Preparation

A total of 6 mM of Zn(NO_3_)_2_·6H_2_O and Co(NO_3_)_2_·6H_2_O was dissolved in a mixed solution of deionized water and ethanol with different proportions under stirring, and then 8 mmol urea and 2 mmol CTAB were added successively with continuous stirring for 30 min. The solution was transferred to a 100 mL Teflon-lined autoclave with pretreated NF and kept at 150 °C for 12 h. After the reaction, the surface of NF was washed with deionized water and ethanol, and it was put into a vacuum drying oven at 60 °C for 8 h to acquire NF with a zinc–cobalt precursor. Then, NF with a ZnCo precursor was put into 0.1 mol/L Na_2_S·9H_2_O aqueous solution at 120 °C for 6 h to obtain NF with ZnCoS. After cooling to room temperature, the ZnCoS was washed with deionized water and ethanol and dried at 60 for 12 h. The sulfurized samples were denoted as ZnCoS-H_2_O, ZnCoS-2:1, ZnCoS-1:1, and ZnCoS-1:2, respectively (2:1, 1:1, and 1:2 are the ratios of deionized water and ethanol, respectively).

### 3.3. Material Characterization

The phase of as-prepared materials was determined by X-ray diffraction (XRD, Ultima IV, wavelength of 1.5418 Å, voltage of 40 kV, current of 40 mA, sweep speed of 5° min). The elemental composition and chemical states of as-prepared materials were characterized by X-ray photoelectron spectroscopy (XPS, Thermo Scientific K-Alpha, San Diego, CA, USA). The high-resolution XPS spectrum was fitted by “Avantage” software (version 5.5). The morphologies and elemental mappings were obtained by field emission–scanning electron microscopy equipped with EDS (FESEM, JEOL JSM-7200F, 15 kV, Tokyo, Japan). The microstructure and lattice fringes were observed by high-resolution transmission electron microscopy (HRTEM, JEOL JEM 2100, 200 kV, Tokyo, Japan).

### 3.4. Electrochemical Measurements

In the three-electrode measurements, ZnCoS samples on nickel foam were used as the working electrode. Nickel foam serves as a current collector due to its high electrical conductivity, good stability in KOH aqueous electrolytes, and three-dimensional porous structure for loading active materials. A Pt plate and an Hg/HgO electrode were used as the counter and reference electrodes, respectively. In the asymmetric supercapacitor test, ZnCoS samples on nickel foam were used as the positive electrode, and activated carbon (AC) pasted on nickel foam was used as the negative electrode. To fabricate the negative electrode, activated carbon, acetylene black and polyvinylidene fluoride (PVDF) were mixed at a mass ratio of 8:1:1, and N-methylpyrrolidone (NMP) was added to make a slurry, which was coated onto a 12 mm diameter nickel foam disc and dried in a vacuum drying oven at 60 °C for 12 h. The 6 M KOH was used as the aqueous electrolyte in all the tests. Cyclic voltammetry (CV) and electrochemical impedance spectroscopy (EIS) measurements were performed on a Modulab XM electrochemical workstation. EIS was tested in the frequency range of 0.01–10^5^ Hz with an amplitude of 5 mV. Galvanostatic charge–discharge (GCD) and cycling stability tests were performed on a land battery test system.

The area specific capacitance Cs (F cm^−2^) of the working electrode is calculated from Equation (1):(6)$$Cs=\frac{I∆t}{A∆V}$$

*I* (A) is the current value in the GCD test, Δ*t* (s) is the discharge time curve, *A* (cm^2^) is the area of the working electrode, and Δ*V* (*V*) is the voltage range of the GCD test. The energy density (*E*, Wh cm^−2^) and power density (*P*, W cm^−2^) of the device are obtained from Equations (2) and (3), respectively:(7)$$E=\frac{C∆V^{2}}{2\times 3600}$$
(8)$$P=\frac{3600E}{∆t}$$

## 4. Conclusions

In summary, bimetallic sulfides of ZnCoS electrode materials were prepared by a solvothermal method, and the morphology of ZnCoS was regulated by adjusting the content of ethanol in the solvent. The XRD results showed that the addition of ethanol reduced the crystallinity of ZnCoS. The SEM observation showed that the microstructure of ZnCoS transformed one-dimensional wires to two-dimensional sheets with increasing amounts of ethanol. Electrochemical tests demonstrated that the ZnCoS-1:2 materials prepared in a solvent with a water/ethanol ratio of 1:2 exhibited the best capacitance of 6.22 F cm^−2^ at a current density of 5 mA cm^−2^, which is superior to that of the most previously reported bimetallic sulfides. The enhanced electrochemical performance could be ascribed to its sheet-assembled spherical structure, which not only shortens the path of ion diffusion but also increases the contact between surface active sites and the electrolyte. Moreover, the spherical structure provides numerous void spaces for buffering the volume expansion and penetration of the electrolyte, which would be favorable for electrochemical reactions. Furthermore, the asymmetric supercapacitors (ASCs) of ZnCoS-1:2//AC were constructed, and they exhibited a maximum energy density of 0.124 mWh cm^−2^ under a power density of 2.1 mW cm^−2^. In addition, even under a high-power density of 21 mW cm^−2^, the energy density can still reach 0.055 mWh cm^−2^. In addition, the device also revealed good cycling stability, and it could retain 68.4% of the initial capacitance after 5000 cycles at a current density of 30 mA cm^−2^.

## Figures and Tables

**Figure 1 molecules-28-06578-f001:**
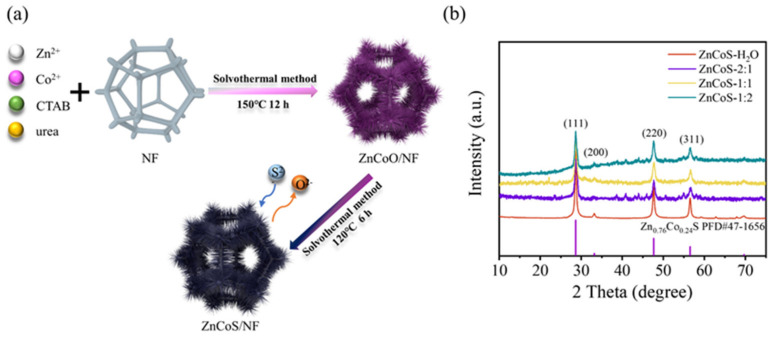
(**a**) Schematic illustration of the growth process of ZnCoS on Ni foam; (**b**) XRD patterns of ZnCoS-H_2_O, ZnCoS-2:1, ZnCoS-1:1, and ZnCoS-1:2.

**Figure 2 molecules-28-06578-f002:**
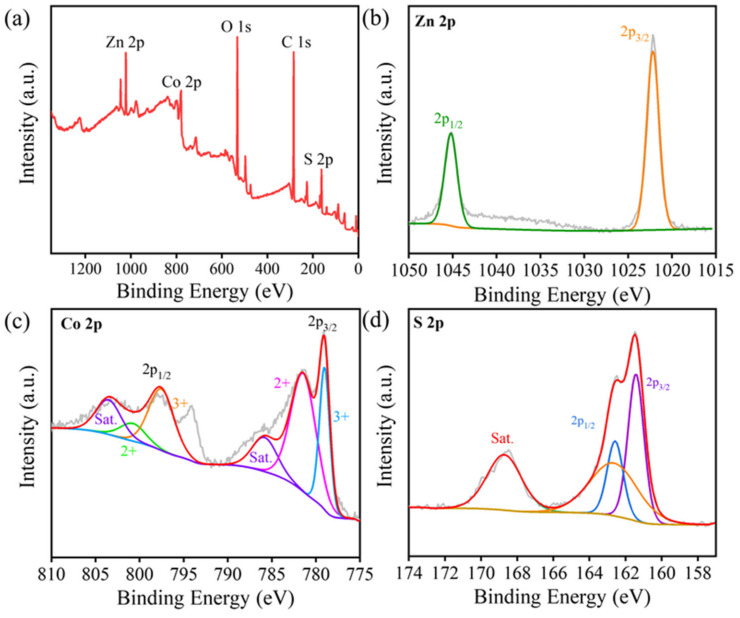
(**a**) XPS full spectra of ZnCoS-1:2, (**b**) Zn 2p spectra of ZnCoS-1:2, (**c**) Co 2p spectra of ZnCoS-1:2, and (**d**) S 2p spectra of ZnCoS-1:2.

**Figure 3 molecules-28-06578-f003:**
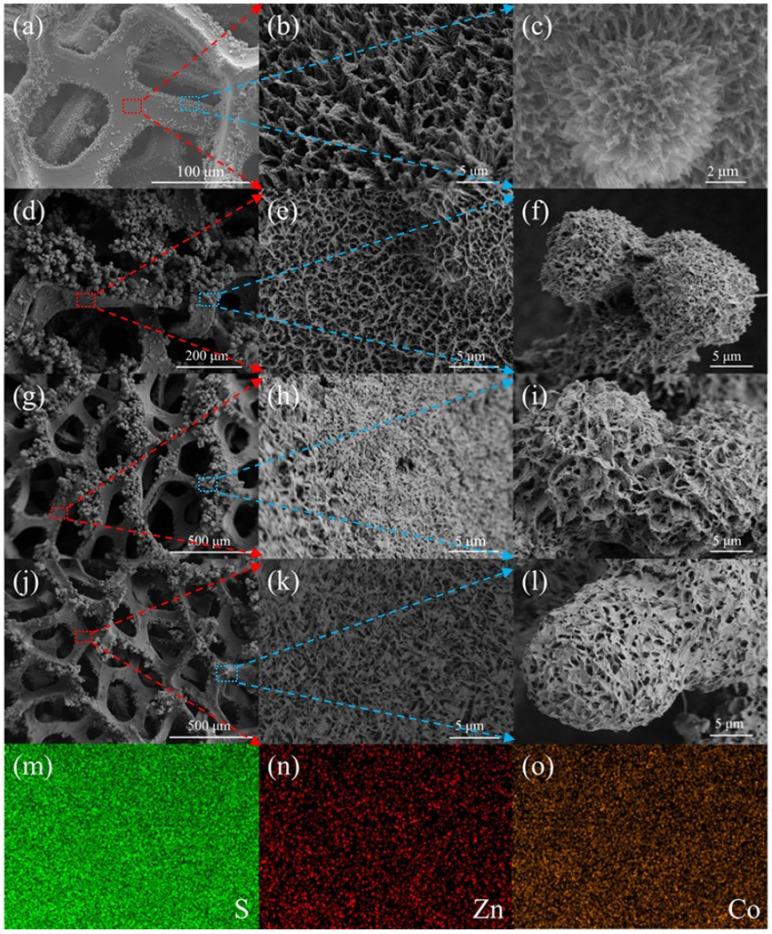
SEM images of (**a**–**c**) ZnCoS-H_2_O, (**d**–**f**) ZnCo-2:1, (**g**–**i**) ZnCo-1:1, and (**j**–**l**) ZnCoS-1:2; (**m**–**o**) S, Zn, and Co elemental mapping of ZnCoS.

**Figure 4 molecules-28-06578-f004:**
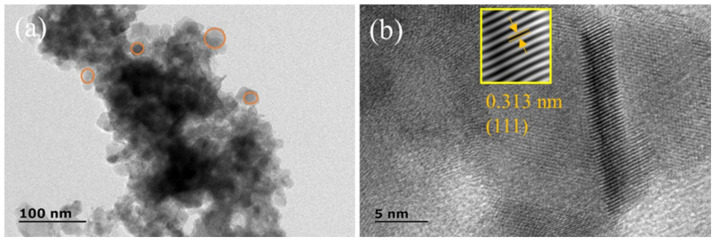
(**a**) TEM image of ZnCoS-1:2, (**b**) HRTEM image of ZnCoS-1:2.

**Figure 5 molecules-28-06578-f005:**
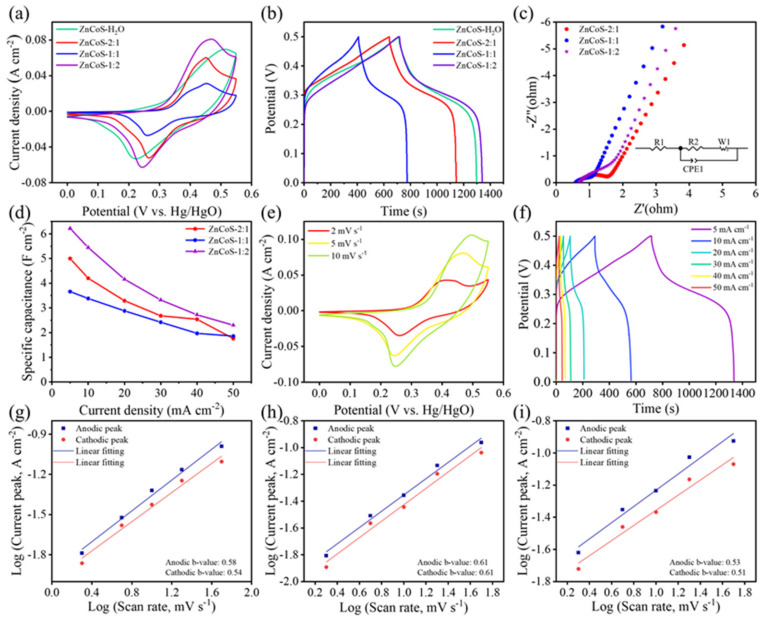
(**a**) CV curves at 5 mV s^−1^ sweep speed of ZnCoS−H_2_O, ZnCoS−2:1, ZnCoS−1:1, and ZnCoS−1:2; (**b**) GCD curves at a 5 mA cm^−2^ of ZnCoS−H_2_O, ZnCoS−2:1, ZnCoS−1:1, and ZnCoS−1:2; (**c**) specific capacitance at 5, 10, 20, 30, 40, 50 mA cm^−2^ of ZnCoS−2:1, ZnCoS−1:1, ZnCoS−1:2; (**d**) EIS plots of ZnCoS−2:1, ZnCoS−1:1, ZnCoS−1:2; (**e**) CV curves of ZnCoS−1:2; (**f**) GCD curves of ZnCoS−1:2. Linear relationship between the log (scan rate) and log (current peak) of (**g**) ZnCoS−2:1, (**h**) ZnCo−1:1, (**i**) ZnCoS−1:2.

**Figure 6 molecules-28-06578-f006:**
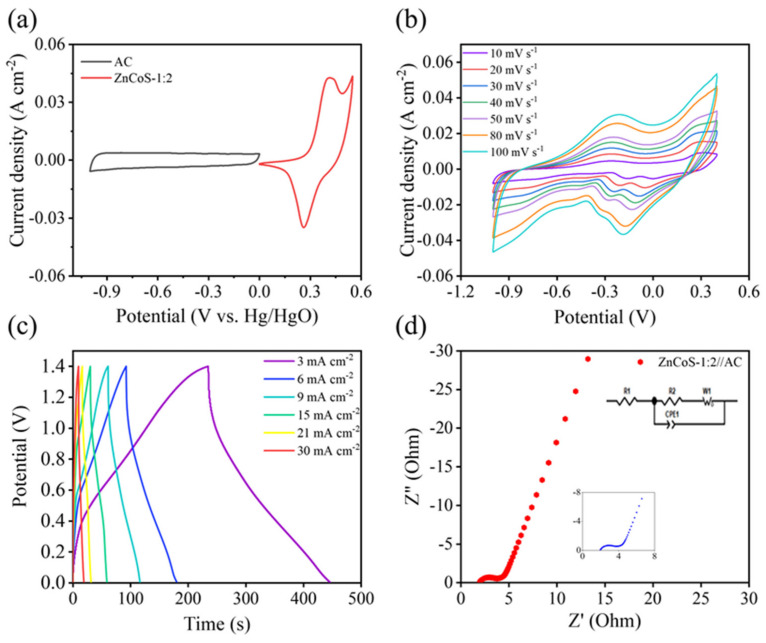
(**a**) CV curves of ZnCoS−1:2 and AC electrodes at a scan rate of 5 mV s^−1^; (**b**) CV curves, (**c**) GCD curves, and (**d**) the Nyquist plot of ZnCoS−1:2//AC ASC.

**Figure 7 molecules-28-06578-f007:**
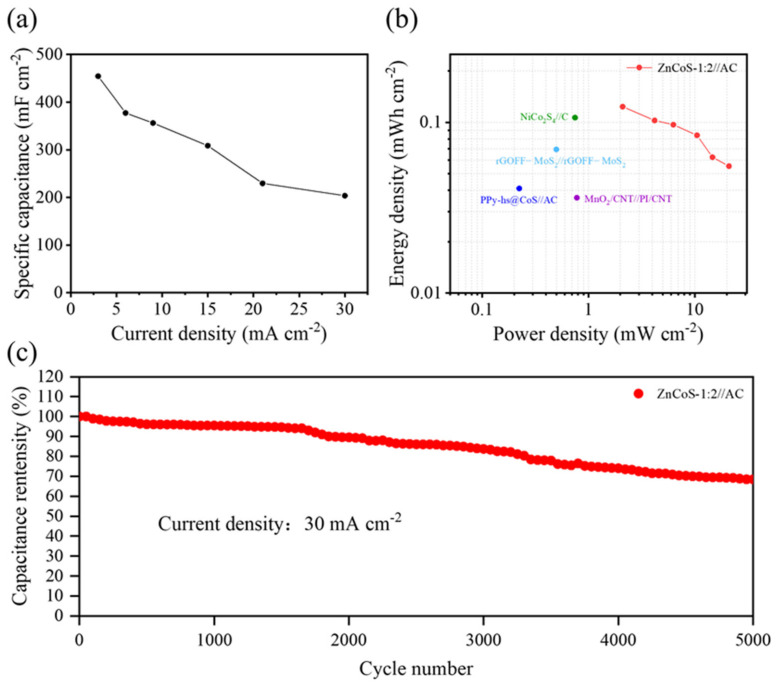
(**a**) Specific capacitance of ZnCoS−1:2//AC ASC under different current densities, (**b**) the Ragone plot, and (**c**) long−-term cycling stability of the ZnCoS−1:2//AC ASC device.

## Data Availability

Data will be made available on request.

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
