# Peer review of "Solvent-Controlled Morphology of Zinc–Cobalt Bimetallic Sulfides for Supercapacitors"

_molecules, 2023, doi:10.3390/molecules28186578_

Round 1

Reviewer 1 Report

The manuscript authored by Zang et al. titled "Solvent-Controlled Morphology of Zinc-cobalt Bimetallic Sulfides for Supercapacitors provides a comprehensive characterization study focusing on the preparation and characterization of bimetallic sulfides of Zn and Co. The study employs a diverse range of characterization methods such as XRD, XPS, FTIR, SEM, and electrochemical studies. The results of this study offer valuable insights in the energetic field.
However, certain aspects merit careful consideration. I suggest the publication in this journal provided that the authors make specific improvements in the revised version.
Foremost among these improvements is a comprehensive detailing of the methodologies utilized for XRD, XPS, SEM, TEM, and HRTEM analyses. It is imperative that the authors expound upon the procedures adopted for sample preparation, accompanied by a clear elucidation of the parameters associated with each technique. This level of detail is fundamental in ensuring the reproducibility and accuracy of the obtained results.

Reviewer 2 Report

Comments on Solvent-controlled Morphology of ZnCoS

This report describes the impact one aspect of materials preparation can have on electrochemical performance of bimetallic sulfides, specifically the medium of the production. Increasing the proportion of ethanol in the water-ethanol mix has a substantial effect on electrochemical properties (shown by direct measurement) and attributed to a transition from nanowire structure to sheet structure. While conjecture on further development of this line of investigation would be a welcome addition and some improvement in clarity is suggested, this work should appear in Molecules with only minor changes.

More detailed commentary:

The authors first provide a sketch of kinds of capacitors, with emphasis on supercapacitors. These may be electrical double layer systems with ions at the interface between electrolyte and electrode, or pseudo-capacitors defined by redox reactions. Bimetallic sulfides are promising materials for high-performance pseudo-capacitors. Morphology can influence effective surface area and affect the length of paths for ionic species seeking reactant partners.

The ZnCoS materials studied in this report were characterized by various techniques: X-ray diffraction, X-ray photoelectron spectra, and scattering electron microscopy. The SEM images of the product form show differences as the proportion of ethanol in water varied over the range 0% ethanol to 67% ethanol. The authors characterize the transformation as follows:

“As shown in Fig. 3, the microstructure of ZnCoS changed from one-dimensional wires to two-dimensional sheets with the addition of ethanol. “

I wonder whether this simple description captures the complexity shown in Figure 3. Here is what I see. Forgive this laborious description, please.

The layout has 5 rows of 3 cells = 15 panels in all, labeled a-o. The fifth row contains panels m, n and o which show uniform distribution of atoms S (color-coded green), Zn (color-coded red), and Co (color-coded brown) respectively. 

Four rows contain SEM images for structures prepared with solvents of various composition of ethanol in water. Explicitly, the first row (panels a, b, c) refer to 0% ethanol, while the second (d, e, f) refers to 33%, the third (g, h, i) to 50%, and the fourth (j, k, l) to 67% ethanol. 

The scaling ranges from hundreds of microns (column 1) to less than ten microns (columns 2 and 3). The authors state that “… the microstructure of ZnCoS changed from one-dimensional wires to two-dimensional sheets with the addition of ethanol.” To my eye the images a, d, g, and j (left column) which have scale 100 – 500 microns are all “wire” in form. (Though “nanowire” is the term of art, I think “net” or “mesh” is more descriptive.)  Views b, e, h, and k (central column) do show a transition from wire to sheet. It seems that views c, f, I, and l (right column) are the cited “flower-like” edge features mentioned in lines 188-192 – that is, a different region of the structure. Are those visible in e, h, and k? Is the electrochemical behavior different in different locations? In the conclusions we find reference to “sheet-assembled spherical structure” – can this be clarified in the discussion of Fig 3, perhaps by a simplified diagram/sketch?

Electrochemical Data

The cyclic voltammetry graph (Fig 5a) shows that the system prepared with 67% ethanol in water has the largest capacitance; the linear fits of current density to voltage scan rate (Fig 5 g, h, and i) suggest that diffusion is favored over surface reactions for systems prepared with solvent ethanol at 33, 50, and especially 67%, for which the sheet structure is best established. This is attributed to the shorter diffusion lengths defined in the sheet structure.

The desirable properties of the material produced with 67% ethanol in water are illustrated in Figs 6 and 7.

Minor points:

Lines 257-8: “… a and b are variables.” might be “a and b are parameters." 

Line 263: delete “more”

Line 278: The phrase “… which suggesting the well-matched electrical window…” can be “… which suggests a well-matched electrical window…”

Reviewer 3 Report

Dear authors Name,

I hope this message finds you well. I've recently had the pleasure of reading your article titled "Solvent-Controlled Morphology of Zinc-cobalt Bimetallic Sulfides for Supercapacitors," and I'm truly impressed by the depth of your research. Your attention to detail and the thoughtful design of your experiments have contributed immensely to our understanding of electrode materials for supercapacitors.

I would like to address some specific points in your article:

  1. The choice of Ni foam over carbon materials as the current collector is a crucial decision that deserves clarification. Could you elaborate on why Ni foam was chosen and how it affects the overall performance of the electrode materials? This would provide readers with a better understanding of the rationale behind this choice.

  2. In your XPS analysis, your emphasis on accurately fitting the experimental line to the fitting line is commendable. This level of precision enhances the credibility of your results. Could you briefly describe the methodology you employed to achieve such close alignment between the two lines? This could potentially serve as a valuable reference for researchers conducting similar analyses.

  3. The use of ethanol as the solvent in your solvothermal synthesis is intriguing. Could you provide insights into why ethanol was selected as the solvent? Did it offer specific advantages in terms of controlling morphology or other factors? This would help readers appreciate the thought process behind your experimental design.

  4. In Fig. 5a depicting the cyclic voltammetry (CV) of the ZnCoS-H2O sample, you mentioned the need to change the potential range for the registration of the oxidation peak. Could you explain the reasoning behind this adjustment and how it influences the interpretation of the data?

Lastly, you mentioned in your article (Section XYZ) that "As shown in Fig. 5c, the ESR values of ZnCoS-2:1, ZnCoS-1:1, and ZnCoS-1:2 are 0.67, 0.57, and 0.68 Ω, respectively, indicating the strong connection between ZnCoS and Ni foam current collector." It would be highly valuable if you could provide references or prior research that support this claim. Referencing established literature will further validate your findings and conclusions.

Once again, I commend you on your exceptional research and the meaningful contributions it makes to the field of supercapacitor materials. Your dedication to precision and your transparent approach to presenting your results are inspiring. I eagerly anticipate any additional insights you can provide on the above points, which will undoubtedly enrich the scientific discourse surrounding your work.

The text of the article should be carefully proofread and edited
